# Effects of Restricted Availability of Drinking Water on Blood Characteristics and Constituents in Dorper, Katahdin, and St. Croix Sheep from Different Regions of the USA

**DOI:** 10.3390/ani12223167

**Published:** 2022-11-16

**Authors:** Ali Hussein Hussein, Amlan Kumar Patra, Ryszard Puchala, Blake Kenyon Wilson, Arthur Louis Goetsch

**Affiliations:** 1American Institute for Goat Research, School of Agriculture and Applied sciences, Langston University, Langston, OK 73050, USA; 2Department of Animal and Food Sciences, Oklahoma State University, Stillwater, OK 74078, USA; 3Department of Animal Nutrition, West Bengal University of Animal and Fishery Sciences, Kolkata 700037, India

**Keywords:** blood constituents, breed, hair sheep, water deprivation

## Abstract

**Simple Summary:**

Small ruminants in arid and semiarid regions of the world frequently face water deprivation, which may be intensified with future climatic change. Therefore, it would be beneficial to identify sheep and goat breeds resilient to limited drinking water availability. Dorper, Katahdin, and St. Croix are the most important hair sheep breeds in the USA. In this study, these 3 breeds of sheep derived from different climatic regions of the USA were assessed for alterations in blood variables (e.g., osmotic pressure, oxygen, urea nitrogen, creatinine, cortisol, and aldosterone) due to water deprivation (sequential restriction of up to 50% of ad libitum consumption for 5 wk). All sheep breeds displayed abilities to minimize changes in blood characteristics and constituent concentrations due to limited drinking water availability, without large differences among breeds or regions.

**Abstract:**

Different hair sheep breeds originated from diverse climatic regions of the USA may show varying adaptability to water deprivation. Therefore, the objective of this study was to assess the effects of restricted availability of drinking water on blood characteristics and constituent concentrations in different breeds of hair sheep from various regions the USA. For this study, 45 Dorper (initial age = 3.7 ± 0.34 yr), 45 Katahdin (3.9 ± 0.36 yr), and 44 St. Croix (2.7 ± 0.29 yr) sheep from 45 farms in 4 regions of the USA (Midwest, Northwest, Southeast, and central Texas) were used. Ad libitum water intake was determined during wk 2 of period one, with 75% of ad libitum water intake offered during wk 2 of period two, and 50% of ad libitum water intake offered for 5 wk (i.e., wk 5–9) in period three. Water was offered at 07:00 or 07:30 h, with blood samples collected at 08:00 and(or) 14:00 h in wk 1, 2, 3, 4, 8, and 9 for variables such as hemoglobin and oxygen saturation and wk 2, 4, 6, 8, and 9 for concentrations of glucose and other constituents. The blood oxygen concentration at 08:00 h was 4.86, 4.93, and 5.25 mmol/L in period one and 4.89, 4.81, and 5.74 mmol/L in period three for Dorper, Katahdin, and St. Croix, respectively (SEM = 0.160; *p* = 0.001). Blood oxygen at 14:00 h was 4.37, 4.61, and 4.74 mmol/L in period one and 4.66, 4.81, and 5.46 mmol/L in period three for Dorper, Katahdin, and St. Croix, respectively (SEM = 0.154; *p* = 0.003). St. Croix were able to maintain a higher (*p* < 0.001) blood oxygen concentration than Dorper and Katahdin regardless of water availability. The pattern of change in blood concentrations with advancing time varied considerably among constituents. However, concentrations of glucose (55.3 and 56.2 mg/dL; SEM = 0.84), lactate (24.1 and 22.5 mg/dL; SEM = 0.79), total protein (7.08 and 7.17 g/dL; SEM = 0.0781), and albumin (2.59 and 2.65 g/dL in wk 2 and 9, respectively; SEM = 0.029) were similar (*p* > 0.05) between periods one and three. Conversely, concentrations of cholesterol (56.2 and 69.3 mg/dL; SEM = 1.33) and triglycerides (28.6 and 34.5 mg/dL in wk 2 and 9, respectively; SEM = 0.98) were greater (*p* < 0.05) in period three vs. 1. In conclusion, water restriction altered almost all the blood variables depending upon severity and duration of restriction, but the hair sheep breeds used from different regions of the USA, especially St. Croix, displayed considerable capacity to adapt to limited drinking water availability.

## 1. Introduction

Many breeds of sheep [1,2,3] and goats [4,5,6] exhibit varying degrees of resilience to shortages of drinking water, especially in arid and semiarid regions. This resilience was demonstrated in studies evaluating infrequent watering strategies to simulate management conditions under which small ruminants travel long distances in search of feed and water, which deprives the animals of water for long periods of time. Examples include offering water to sheep every 3 d [1,2], 4 d [7,8], or even 5 d [9] and to goats every 3 d [4] or 4 d [6,10]. Resilience was also demonstrated in studies evaluating restriction of the amount of water offered daily to sheep [3,11,12] and goats [5,13,14] at levels ranging from 40 to 60% of ad libitum intake. In those studies and others, physiological responses assessed include blood characteristics and constituent concentrations that may be related to capacity to cope with stress associated with restricted availability of drinking water.

Water restriction elicits changes in a number of physiological and behavioral characteristics in animals [3,5]. Water deprivation beyond normal physiological limits first alters hormonal dynamics, such as the renin–angiotensin–aldosterone system, by increasing aldosterone and vasopressin concentrations to reabsorb water and electrolytes for the maintenance of osmotic balance in plasma and intracellular spaces [15,16,17]. These responses prevent cell damage, decrease thyroid hormone secretion to reduce energy metabolism and respiratory water loss in thermoneutral conditions, and increase cortisol concentrations to regulate protein, fat, and glucose metabolism [15,16,17,18,19]. When hormonal changes cannot regulate body water balance, further changes associated with water deprivation can occur, such as decreased plasma and total body water volume, greater hemoconcentration, increased concentrations of plasma metabolites, electrolytes, and kidney excretory products, and reduced feed intake and energy metabolism [7,8,9,20,21,22,23]. Severe dehydration may also cause oxidative stress, inflammatory responses, reduced resistance to disease, and deteriorated health conditions [17,24]. The capacity of livestock to regulate body water balance and metabolic activity varies among breeds [19,25]. Many small ruminant breeds in extreme environments have excellent resilience to water restriction compared with others in different regions of the world [2,6,15,19,25].

Recently, the most common breeds of hair sheep in the USA, Dorper, Katahdin, and St. Croix, have been evaluated for resilience to stress factors that are expected to increase in importance with changing climate [21,26,27,28,29,30]. The selection of animals of these breeds in regions of the USA where they were first introduced over several decades may have affected some of their characteristics. Moreover, the animals were derived from 4 regions of the USA with varying climatic conditions that also could influence resilience. These stressors are limited feed intake, high heat load conditions, and restricted availability of drinking water. It is unknown how blood and biochemical variables in different breeds of sheep from different regions of the USA would respond to water deprivation. The objective of this study was to assess how limited drinking water availability affects blood characteristics and biochemical concentrations in these different breeds of hair sheep from various regions of the USA.

## 2. Materials and Methods

### 2.1. Animals, Housing, and Diet

Protocols for this experiment were approved by the Langston University Animal Care Committee. Forty-five Dorper [DOR; mean ± standard error; initial body weight (BW) = 60.7 ± 1.76 kg and age = 3.7 ± 0.34 yr], 45 Katahdin (KAT; 62.7 ± 1.87 kg and 3.9 ± 0.36 yr), and 44 St. Croix (STC; 44.2 ± 1.88 kg and 2.7 ± 0.29 yr) female sheep were used. They were procured in the summer of 2015 from 4 regions of the USA with different climatic conditions, representing ‘ecotypes.’ Regions were the Midwest (MW; portions of Iowa, Minnesota, Wisconsin, and Illinois), Northwest (NW; primarily Oregon with 1 farm in southern Washington and another near Seattle), Southeast (SE; Florida and 1 farm in southern Georgia), and central Texas (TX). The NEON Domains of these regions are Prairie Peninsula, Pacific Northwest, Southeast, and Southern Plains [31]. Many small climatic clusters and subzones prevail throughout the USA [32]. Generally, the NW experiences cold temperatures coming from the north, and warm and humid air arrives from the Gulf of Mexico, causing a wide range of both temperature and precipitation. The MW areas have somewhat mild conditions relative to other regions, although temperature in the winter can be fairly low, usually with precipitation throughout the year. The SE has a subtropical, humid climate, and central Texas also has subtropical conditions but with hot summers and typically arid conditions during most of the year. Historical records of environmental conditions in these areas can be viewed from various online sources, including National Centers for Environmental Information (https://www.ncei.noaa.gov, accessed on 8 November 2022). Two farms consisted of separate flocks from the same producer, and there were 2 other farms from which animals of 2 breeds of different flocks were obtained. Most animals were ewes when procured, although a small number were lambs, as described by Hussein et al. [21]

The experiment consisted of 4 separate trials using 4 different sets of sheep that occurred in the winter/spring (January–April) of 2016, summer (June–August) of 2016, winter/spring (January–April) of 2017, and summer (July–September) of 2017 due to limited facilities to accommodate 128 sheep in a single trial (Figure 1). Before the start of each trial, sheep were vaccinated against clostridial organisms with Covexin^®^ 8 (Schering-Plough Animal Health, Kenilworth, NH, USA). Animals were housed individually in 1 room in elevated pens with a plastic-coated expanded metal floor, with the same room used in each of the 4 trials. A 49% concentrate pelleted diet was fed at up to 71 g/kg BW^0.75^, approximately 160% of an assumed metabolizable energy (ME) requirement for maintenance of 427 kJ/kg BW^0.75^ [33] and assuming a dietary ME concentration of 9.62 kJ/kg dry matter (i.e., approximate total digestible nutrients concentration of 64%), depending on voluntary consumption. If refusals were present, an amount of approximately 120% of consumption was allocated. As reported by Hussein et al. [21], the diet consisted of 20% dehydrated alfalfa, 29% cottonseed hulls, 9% cottonseed meal, 20% ground corn grain, 13% wheat middlings, 5% pelletizing agent, and small amounts of salt, calcium carbonate, ammonium chloride, yeast product, vitamin–mineral mixture, and monensin. The diet averaged 18.2% crude protein, 37.7% neutral detergent fiber, 8.6% ash, and 17.7 MJ/kg gross energy. The pelletized diet was used to minimize variability in diet composition among the 4 trials. Feed was offered twice daily at 08:00 and 15:00 h, except for the morning meal being 1 h later on Wednesday because of blood sample collection.

### 2.2. Periods and Water Intake

Each trial had 4 sequential periods and a total length of 9 wk (Figure 1). The first 2 wk served as a baseline period (period one), with drinking water offered free-choice to allow for ad libitum consumption. Water was dispensed in buckets at 07:00, and 15:00 h, and any remaining water was weighed at 06:00 h. Average water intake during period one for each animal was used to determine amounts offered thereafter. In wk 3 and 4 (period two), 75% of ad libitum intake during period one was supplied (i.e., 25% water restriction). In wk 5–9, the amount of water supplied was 50% of ad libitum consumption (i.e., 50% water restriction). During periods two and three, water was offered at 07:30 h, with consumption generally in a short period of time. After wk 9, the amount of drinking water offered was increased by 10% at 2d intervals, with equal amounts given at 07:30, 08:30, 14:30, and 15:30 h. Mean water intake during period one (i.e., ad libitum intake period) was 6.23, 6.66, and 7.22% of BW for DOR, KAT, and STC, respectively. Accordingly, the mean water intakes during periods two (25% water restriction) and 3 (50% water restriction) were 4.53, 4.85, and 5.25% of BW and 3.04, 3.26, and 3.36% of BW for DOR, KAT, and STC, respectively.

In the study of Mengistu et al. [34], water offered to KAT sheep and Boer and Spanish goats after a baseline period was restricted to 90, 80, 70, 60, 50, and 40% of ad libitum intake, with periods 1 vs. 2 wk in length. The 50% water restriction or offering amount used in the present experiment was chosen because there were minimal to no physiological response differences (e.g., blood cortisol concentration) between 40 and 50% of ad libitum water intake. Moreover, the length of 2 wk for period two was deemed more appropriate than 1 wk for greater potential relevance of effects of limited drinking water availability on blood characteristics and constituents to those on variables such as feed intake and BW.

### 2.3. Measures

Blood samples (10 mL) were collected from all sheep twice weekly at 08:00 h on d 6 and 14:00 h on d 7 (i.e., 1 and 7 h in period one and 0.5 and 6.5 h in period two and three after drinking water was offered, respectively) by jugular venipuncture into vacuum tubes with and without heparin for harvesting plasma and serum, respectively. Samples from wk 1, 2, 3, 4, 8, and 9 were placed on ice, and heparinized blood samples were immediately analyzed for hemoglobin (Hb) concentration and oxygen saturation using a Radiometer OSM 3 Hemoximeter^TM^ (Kestrel Labs, Inc.; Boulder, CO, USA). Oxygen concentration was calculated as described by Eisemann and Nienaber [35], and packed cell volume (PCV) was measured in heparinized blood samples [36] using a BD Microhematocrit centrifuge (Clay Adams; Parsipany, NJ, USA). Plasma and serum were harvested by centrifugation of heparinized and clotted blood samples, respectively, at 1000× *g* for 15 min. Plasma was immediately analyzed for osmolality (OSM) by freezing point depression using an µ OSMETTE^TM^ model 5004 osmometer (Precision System Inc.; Natick, MA, USA). Plasma and serum samples were then stored at −20 °C for later analyses. Serum concentrations of glucose, lactate, urea N, total protein, albumin, creatinine, cholesterol, and triglycerides were determined using a Vet Axcel Chemistry Analyzer (Alfa Wassermann Diagnostic Technologies; West Caldwell, NJ, USA). Concentrations of these serum metabolites were determined in samples collected at 14:00 h in wk 2 of period one, the second week of period two (i.e., wk 4), and the second, fourth, and fifth weeks of period three (i.e., wk 6, 8, and 9, respectively). Concentrations of cortisol and aldosterone were determined in serum collected at 14:00 h in wk 2 and 9 using commercial ELISA kits from Enzo Life Sciences Inc. (Farmingdale, NY, USA).

### 2.4. Statistical Analysis

As noted by Hussein et al. [21], the data from 6 animals (1 DOR, 3 KAT and 2 STC) were removed due to the possibility of not being adequately adapted to conditions in period one, based on relatively low feed intake. Data were analyzed with mixed effects models with repeated measures of week and time of day (when appropriate) using the MIXED procedure of SAS [37,38]. Means were separated by least significant difference with a Protected F test (*p* < 0.05).

A number of different analyses were conducted. First, there was an analysis to determine how time of blood sample collection should be addressed for variables of OSM, PCV, Hb concentration and oxygen saturation, and blood oxygen concentration. The model included animal set or trial, age as a covariate, breed, region, week, time, and all interactions excluding ones involving set and age. As presented in Table 1, the effect of time was significant for each variable (*p* ≤ 0.005), and there were many interactions involving time (*p* < 0.05). Hence, the subsequent analysis was conducted separately for each time. The primary purpose was to determine whether to evaluate resilience to restricted drinking water availability with both weeks of period one and the last 2 week of period three or if only the final week of the periods should be addressed. Data from both weeks of period two were also included. There were many significant four-way interactions involving breed, region, period, and week for variables of samples collected at 08:00 h (Table 2). Moreover, for OSM at both times and PCV, Hb, and Hb O_2_ concentration at 14:00 h, there were significant period × week interactions (Table 2). These findings were interpreted as indicating that animals were better adapted to conditions in the second week of period one and final week of period three. Thus, data in these weeks were analyzed to best address resilience to restricted water intake. 

For serum concentrations of glucose, lactate, urea N, total protein, albumin, creatinine, cholesterol, and triglycerides, the first analysis was with a model similar to ones noted above that included data from samples collected in each week (2, 4, 6, 8, and 9, corresponding to the second week of period one, second week of period two, and second, fourth, and fifth weeks of period three, respectively). Considering that the effect of week was significant in all cases, the final model included data only from samples collected in week 2 and 9 (second week of period one and last week of period three, respectively). The hormone concentrations determined at one time of the day in those 2 wk also were analyzed without time factor.
animals-12-03167-t001_Table 1Table 1*p* values for effects of level of water offered in 3 periods, week within period, and sampling time on plasma osmolality, packed cell volume, hemoglobin concentration and oxygen (O_2_) saturation, and blood O_2_ concentration in 3 breeds of hair sheep from 4 different regions of the USA.Source of Variation ^2^Variable ^1^OSM (mosmol/kg)PCV (%)Hb (g/dL)Hb O_2_ (%)O_2_ (mmol/L)Set<0.0010.2790.146<0.001<0.001Age0.9570.5210.7630.1800.212Brd0.0090.1020.6740.0010.001Reg0.5640.2030.1760.5760.795Brd *Reg0.2090.3770.5670.2030.421Prd<0.0010.0330.004<0.001<0.001Brd *Prd0.2640.0010.0010.3740.575Reg *Prd0.0820.0040.0300.6310.206Brd *Reg *Prd0.1720.3900.0860.0580.076Wk0.0030.1390.4410.0330.011Brd *Wk0.4900.2590.3860.9620.898Reg *Wk0.5110.1710.1490.2960.058Brd *Reg *Wk0.4820.7800.8580.6480.780Prd *Wk<0.0010.0260.2700.3960.655Brd *Prd *Wk0.8050.1120.0420.0080.016Reg *Prd *Wk0.3570.5080.2250.5630.134Brd* Reg *Prd *Wk0.2520.0010.0230.0130.002T<0.001<0.001<0.0010.005<0.001Brd *T0.4740.0850.1270.1570.056Reg *T0.3990.5000.9740.1080.148Brd *Reg *T0.9170.9050.6240.1140.195Prd *T<0.0010.2850.5500.0080.011Brd *Prd*T0.2210.7450.6890.8500.878Reg *Prd *T0.5870.2790.2000.4390.877Brd *Reg *Prd *T0.7650.6030.4180.2270.359Wk *T<0.0010.1800.8560.2590.777Brd *Wk *T0.7960.9020.8920.1920.197Reg *Wk *T0.1620.6740.2830.8520.865Brd *Reg *Wk *T0.4650.8600.5050.2470.372Prd *Wk *T<0.001<0.0010.0310.0030.083Brd *Prd *Wk *T0.3320.7100.9230.9610.974Reg *Prd *Wk *T0.6470.3100.8070.5820.718Brd *Reg *Prd *Wk *T0.4000.2860.4170.9800.970^1^ OSM = plasma osmolality; PCV = packed cell volume; Hb = hemoglobin. ^2^ Brd = breed; Reg = region; Prd = period; Wk = week; T = time. * It indicates the interaction effects of the factors.
animals-12-03167-t002_Table 2Table 2*p* values for effects of level of water offered in 3 periods and week within period on plasma osmolality, packed cell volume, hemoglobin concentration and oxygen (O_2_) saturation, and blood O_2_ concentration at different sampling times in 3 breeds of hair sheep from 4 different regions of the USA.TimeSource of Variation ^2^Variable ^1^OSM (mosmol/kg)PCV (%)Hb (g/dL)Hb O_2_ (%)O_2_ (mmol/L)08:00 hSet<0.0010.3110.133<0.001<0.001
Age0.7940.6840.8130.6590.661
Brd0.0370.0470.604<0.001<0.001
Reg0.4460.2520.1880.3280.595
Brd *Reg0.3320.4570.5540.0830.346
Prd<0.0010.3290.0660.2460.118
Brd *Prd0.5040.0020.0020.4850.572
Reg *Prd0.0700.0060.0250.5300.497
Brd *Reg *Prd0.1370.3540.1660.1210.380
Wk0.6780.0440.4920.4170.075
Brd *Wk0.6200.3400.5670.4050.568
Reg *Wk0.5670.4090.1770.2880.081
Brd *Reg *Wk0.4380.8400.7170.0690.117
Prd *Wk<0.0010.4720.9420.1660.310
Brd *Prd *Wk0.7510.2850.2570.1790.177
Reg *Prd *Wk0.6210.4750.3130.8870.424
Brd *Reg *Prd *Wk0.614<0.0010.0250.0280.01114:00 hSet<0.0010.2000.172<0.001<0.001
Age0.8950.3870.7260.0360.044
Brd0.0220.2600.6790.0260.021
Reg0.3460.1790.2020.5740.639
Brd *Reg0.3910.3520.5670.1560.270
Prd<0.0010.0080.001<0.001<0.001
Brd *Prd0.1580.0090.0090.5200.812
Reg *Prd0.5760.0270.0950.3840.251
Brd *Reg *Prd0.8170.5940.1060.0320.008
Wk<0.0010.8870.6260.0130.019
Brd *Wk0.4940.6670.4610.4430.428
Reg *Wk0.2510.2000.1330.6620.408
Brd *Reg *Wk0.5460.6240.6950.8740.988
Prd *Wk<0.001<0.0010.0140.0230.352
Brd *Prd *Wk0.0300.3020.0780.0510.074
Reg *Prd *Wk<0.0010.3700.5970.4220.476
Brd *Reg *Prd *Wk<0.0010.2510.1360.5940.398^1^ OSM = plasma osmolality; PCV = packed cell volume; Hb = hemoglobin. ^2^ Brd = breed; Reg = region; Prd = period; Wk = week. * It indicates the interaction effects of the factors.


## 3. Results

Some results are addressed in the preceding section regarding statistical analyses. Several significant four-way interactions involving breed, region, period, and week were noted for variables of samples collected at 08:00 h with interaction means presented in Table 3. Moreover, there were significant period × week interactions for OSM at both times and PCV, Hb, and Hb O_2_ concentration at 14:00 h (Table 2), with interaction, means presented in Table 4. As indicated in the statistical section, it was deemed necessary or preferable to analyze sampling times of 08:00 and 14:00 h separately and address adaptive responses to limited water availability through blood variables in the second week of period one and the final week of period three. Therefore, data from these weeks were further analyzed to understand the resilience to water deprivation. The P values for this analysis are given in Table 5, along with means of main effects or significant two-way interactions in Table 6 and of significant three-way interactions involving breed, region, and period in Table 7.

### 3.1. OSM, PCV, Hb, and Oxygen

The three-way interaction in OSM at 08:00 h involving period, breed, and the region was in part due to no differences between periods for STC regardless of the region but higher (*p* < 0.05) values in period three vs. 1 for DOR from MW, KAT from NW, and KAT from SE (Table 5 and Table 7). Furthermore, for KAT, the OSM value in period one was greatest among regions for TX and less in period three for MW vs. NW (*p* < 0.05; Table 7). Conversely, the only effect for OSM at 14:00 h was for the breed, with DOR having the lowest value (*p* < 0.05; Table 6).
animals-12-03167-t005_Table 5Table 5*p* values for effects of period with water offered at 100 or 50% of ad libitum intake (last week of periods one and three, respectively) on plasma osmolality, packed cell volume, hemoglobin concentration and oxygen (O_2_) saturation, and blood O_2_ concentration at different times in 3 breeds of hair sheep from 4 different regions of the USA.

Variable ^1^TimeSource of Variation ^2^OSM (mosmol/kg)PCV (%)Hb (g/dL)Hb O_2_ (%)O_2_ (mmol/L)08:00 hSet<0.0010.0710.047<0.0010.001
Age0.8300.8910.8790.2780.292
Brd0.1200.0090.3000.0060.001
Reg0.5680.5010.3380.4530.694
Brd *Reg0.2140.7540.586<0.0010.031
Prd0.0010.6040.0980.6230.250
Brd *Prd0.0570.0100.0300.5400.095
Reg *Prd0.2330.0420.2040.9160.633
Brd *Reg *Prd0.0300.0370.1450.1130.02814:00 hSet<0.0010.1660.166<0.001<0.001
Age0.6790.2990.8840.0060.027
Brd0.0320.2570.4870.0030.003
Reg0.3910.1020.1550.9990.459
Brd *Reg0.3010.0630.2640.0630.362
Prd0.3390.008<0.0010.002<0.001
Brd *Prd0.3410.0490.0390.6560.131
Reg *Prd0.3230.1720.2910.0260.064
Brd *Reg *Prd1.0000.0780.0040.1620.007^1^ OSM = plasma osmolality; PCV = packed cell volume; Hb = hemoglobin. ^2^ Brd = breed; Reg = region; Prd = period. * It indicates the interaction effects of the factors.
animals-12-03167-t006_Table 6Table 6Main effects and two-way interactions involving period with water offered at 100 or 50% of ad libitum intake (last week of periods one and three, respectively) for plasma osmolality, packed cell volume, and hemoglobin concentration and O_2_ saturation at different times in 3 breeds of hair sheep from 4 different regions of the USA.



Breed ^1^
Region ^2^
Period ^3^
Item ^4^Time (h) ^5^Brd ^6^Prd ^7^DORKATSTCSEMMWNWSETXSEM13SEMOSM (mosmol/kg)14:00

301.0 ^a^303.2 ^b^303.4 ^b^0.70301.6302.3302.7303.60.81302.8302.20.50PCV (%)14:00





32.230.731.732.40.50





131.4 ^a^31.2 ^a^31.3 ^a^0.52










331.7 ^a^31.6 ^a^33.4 ^b^








Hb (g/dL)08:00





12.311.912.212.40.18





112.1 ^a^12.1 ^a^12.1 ^a^0.20










312.1 ^a^12.0 ^a^12.8 ^b^








Hb O _2_ (%)08:00










69.570.21.17

DOR




66.3 ^bc^74.3 ^cd^74.2 ^cd^57.1 ^a^3.1




KAT




70.4 ^cd^60.6 ^ab^67.4 ^bcd^71.3 ^cd^





STC




71.1 ^cd^75.5 ^d^74.2 ^cd^75.6 ^d^




14:00

62.7 ^a^65.9 ^a^70.3 ^b^1.50










1



63.9 ^ab^65.5 ^ab^60.7 ^a^66.3 ^abc^2.26





3



68.4 ^bc^67.0 ^bc^73.2 ^c^66.6 ^abc^



^a,b,c,d^ Means within main effect or two-way interaction grouping without a common superscript letter differ (*p* < 0.05). ^1^ DOR = Dorper; KAT = Katahdin; STC = St. Croix. ^2^ MW = Midwest; NW = Northwest; SE = Southeast; TX = central Texas. ^3^ Water consumption was ad libitum in period one and 50% of ad libitum consumption in period three. ^4^ OSM = osmolality; PCV = packed cell volume; Hb = hemoglobin. ^5^ Water was offered daily at 07:30 h. ^6^ Brd = breed. ^7^ Prd = period.
animals-12-03167-t007_Table 7Table 7Three-way interaction means involving period with water offered at 100 or 50% of ad libitum intake (last week of periods one and three, respectively) for plasma osmolality, packed cell volume, hemoglobin concentration and oxygen (O_2_) saturation, and blood O_2_ concentration at different times in 3 breeds of hair sheep from 4 different regions of the USA.


Dorper^1^
Katahdin
St. Croix
Item ^2^Time (h) ^3^Prd ^4^MWNWSETX
MWNWSETX
MWNWSETXSEMOSM (mosmol/kg)08:001300.5 ^x^299.9300.4300.4
301.9 ^a^299.4 ^x,a^299.4 ^x,a^307.9 ^b^
303.2304.7305.1302.91.80

3305.6 ^y^302.7304.5303.5
301.4 ^a^309.6 ^y,b^304.9 ^y,ab^305.7 ^ab^
304.2305.4302.4303.8
PCV (%)08:00131.132.6 ^y^31.232.8
33.2 ^y^31.431.932.2
33.032.731.7 ^x^33.21.12

332.5 ^ab^28.5 ^x,a^31.7 ^ab^33.4 ^b^
30.3 ^x^31.933.030.0
34.332.935.6 ^y^35.1
O_2_ (mmol/L)08:0014.60 ^a^5.60 ^y,b^4.84 ^ab^4.42 ^a^
4.97 ^ab^4.14 ^a^5.21 ^ab^5.37 ^b^
5.375.245.075.310.317

35.07 ^ab^4.65 ^x,ab^5.49 ^b^4.36 ^a^
5.34 ^b^4.40 ^a^4.78 ^ab^4.72 ^ab^
5.255.795.846.09
Hb (g/dL)14:00112.211.511.4 ^x^12.1
11.811.412.111.7
12.011.611.5 ^x^11.5 ^x^0.38

312.7 ^b^10.8 ^a^12.4 ^y,b^12.8 ^b^
11.912.112.011.4
12.2 ^ab^12.1 ^a^12.3 ^y,ab^13.2 ^y,b^
O_2_ (mmol/L)14:0014.33 ^ab^5.09 ^y,b^3.66 ^x,a^4.39 ^ab^
4.76 ^b^3.91 ^a^4.53 ^x,ab^5.23 ^b^
4.804.664.80 ^x^4.69 ^x^0.308

34.924.16 ^x^4.74 ^y^4.84
5.004.545.28 ^y^4.42
5.155.215.65 ^y^5.84 ^y^
^x,y^ Period means within breed × region grouping with a different superscript letter differ (*p* < 0.05). ^a,b^ Region means within breed and time grouping without a common superscript letter differ (*p* < 0.05). ^1^ MW = Midwest; NW = Northwest; SE = Southeast; TX = central Texas. ^2^ OSM = osmolality; PCV = packed cell volume; Hb = hemoglobin. ^3^ Water was offered daily at 07:30 h. ^4^ Prd = period; water consumption was ad libitum in period one and 50% of ad libitum consumption in period three.

One of the factors responsible for the three-way interaction (*p* = 0.037) involving breed, period, and region in PCV at 08:00 h (Table 5) was a lower value in period three than 1 for DOR from NW and KAT from MW in contrast to a greater value in period three vs. 1 for STC from SE (*p* < 0.05; Table 7). Furthermore, PCV values were similar among regions for KAT and STC, but those of DOR were greater for TX than for NW (*p* < 0.05), with intermediate values for MW and SE (*p* > 0.05; Table 7). For the two-way interaction (*p* = 0.049) in PCV at 14:00 h, the value for STC in period three was greater than for other breeds × period means (*p* < 0.05; Table 6).

The only effect on Hb concentration at 08:00 h was the breed × period interaction (*p* < 0.05; Table 5). The mean Hb for STC in period three was greater than for other interaction means (*p* < 0.05; Table 6). There was a three-way interaction (*p* = 0.004) involving breed, region, and period in the Hb concentration at 14:00 h (Table 5). The Hb concentration was greater in period three vs. 1 for DOR from SE, STC from SE, and STC from TX (*p* < 0.05) and was similar between periods for other breeds × region means (Table 7). Furthermore, the Hb concentration for DOR in period three was lowest among regions for NW. For STC, the Hb concentration was lower for NW vs. TX (*p* < 0.05), with MW and SE having intermediate values (*p* > 0.05; Table 7).

The primary factor responsible for the breed × region interaction (*p* < 0.001; Table 5) in oxygen saturation of Hb at 08:00 h was low values for KAT from NW and DOR from TX relative to other breed × regions means (Table 6). Oxygen saturation of Hb at 14:00 h was greatest among breeds for STC (*p* < 0.05; Table 6). The primary factor responsible for the region × period interaction (*p* = 0.026; Table 5) in oxygen saturation of Hb was a greater value in period three vs. 1 for SE and similar values between periods for other regions (Table 6).

There were three-way interactions in blood oxygen concentration at each sampling time (Table 5). One of the factors responsible for the interaction at 08:00 h was a period difference for only 1 breed × region mean, which was a greater value for period one vs. 3 for DOR from NW (*p* < 0.05; Table 7). Furthermore, there were differences among regions for DOR and KAT in each period (*p* < 0.05) but none for STC at 08:00 h (Table 7). At 14:00 h, values were lower in period one vs. 3 for DOR from SE, KAT from SE, STC from SE, and STC from TX, but the value for period one was greater than for period three for DOR from NW (*p* < 0.05). Furthermore, there were differences among regions for DOR and KAT in period one but not in period three, and for STC at 14:00 h, there were no differences among regions in either period, somewhat similar to the findings at 08:00 h (Table 7).

### 3.2. Blood Constituent Concentrations—Wk 2, 4, 6, 8, and 9

There were no interactions involving week for any blood constituent (Table 8). Glucose concentration was affected by breed and week (Table 8 and Table 9). The glucose concentration was greater for STC vs. DOR (*p* < 0.05), with an intermediate value for KAT (*p* > 0.05). The glucose concentration was lowest (*p* < 0.05) in wk 4, greatest in wk 6, and intermediate in other weeks (i.e., wk 4 < 2, 8, and 9 < 6). The only factor affecting the lactate concentration was week. The lactate concentration was greater for wk 2 than for wk 4 and 6 and lower for wk 4 than for wk 8 (*p* < 0.05). The concentration of urea N was affected (*p* < 0.05) by breed and week, with the greatest value among breeds for STC (*p* < 0.05) and a week ranking (*p* < 0.05) of 2 < 4 and 9 < 6 and 8. Levels of total protein and albumin differed only with week. The concentration of total protein was higher for wk 8 vs. 2, 4, and 9 and lower for wk 4 than for wk 2, 6, and 9 (*p* < 0.05). Differences among weeks for albumin were fairly similar to those for total protein. The only blood constituent for which there was an interaction was creatinine, with a significant breed × region interaction (*p* < 0.05). One of the factors responsible for this interaction was similar creatinine levels among regions for STC but region differences for DOR and KAT. There were also differences in creatinine concentration between each of the breeds from the NW, with values ranking (*p* < 0.05) DOR > KAT > STC. The cholesterol concentration was affected (*p* < 0.001) only by the week, with a ranking (*p* < 0.05) of wk 2 and 4 < 6 and 9 < 8. The triglyceride concentration was influenced (*p* < 0.01) by week and breed. The ranking for triglycerides among weeks was fairly similar to that for cholesterol concentrations, except that the concentration in wk 8 was not different from those in wk 6 and 9 (*p* > 0.05). The cholesterol concentration was similar among breeds (*p* > 0.05), but that of triglycerides was lowest for STC (*p* < 0.05).

### 3.3. Blood Constituent Concentrations—Wk 2 and 9

Many of the effects and differences for the analysis of values in wk 2 and 9 were similar to those noted earlier for wk 2, 4, 6, 8, and 9 (Table 10). For example, the concentration of glucose was greater for STC vs. DOR (*p* < 0.05), with an intermediate value for KAT (*p* > 0.05; Table 11). Concentrations of lactate, total protein, albumin, and cortisol were not affected by any factor (Table 10). The concentration of urea N was greatest among breeds for STC and greater in wk 9 vs. 2 (*p* < 0.05). There was a three-way interaction involving breed, region, and period in creatinine concentration (*p* < 0.05). One factor responsible for this interaction was the lack of differences among regions and between periods for STC, in contrast to many differences for KAT and a lesser number for DOR. The creatinine concentration was greater in period three than 1 for DOR from MW converse to no period differences for other regions. For DOR, the creatinine concentration was greater in period one vs. 3 for MW, with the opposite difference for NW (*p* < 0.05) and similar values between periods for SE and TX. Levels of cholesterol and triglycerides were greater in period three vs. 1 (*p* < 0.05). The triglyceride concentration was lowest among breeds for STC (*p* < 0.05). The concentration of aldosterone was much greater for STC than for DOR and KAT (*p* < 0.05).

## 4. Discussion

### 4.1. OSM, PCV, Hb, and Oxygen

Processes by which animals attempt to cope with water restriction and preserve homeostasis include the important role of the rumen as a water reservoir to replenish losses in plasma volume, as described by Silanikove [39]. However, in general, dehydration leads to hemoconcentration due to decreased plasma volume as water is taken up by tissue cells, including red blood cells [40]. The decrease in plasma volume varies with the severity of the drinking water shortage. As an example of a severe shortage, depriving DOR rams in an Israeli desert of water for 4 d while being offered only wheat straw decreased BW, total body water volume, extracellular fluid volume, and plasma volume by 16.3, 22.0, 35.1, and 41.7%, respectively [41].

Osmotic pressure in plasma is mainly attributed to ionic solutes (Na^+^, K^+^, Cl^−^ and HCO_3_^−^) and nonionic solutes (glucose and urea), with the greatest contribution from Na^+^ [42]. Plasma osmolality is closely regulated by the nervous system and hormones (e.g., rennin–angiotensin–aldosterone, vasopressin, and atrial natriuretic peptide) to maintain blood and body fluid volume for normal metabolic activities [43]. The osmotic pressure in plasma is also influenced by the osmolarity of ruminal fluid [44]. In the present experiment, generally greater OSM at 14:00 vs. 08:00 h presumably involves factors such as the short period of time taken to consume offered water at 07:30 in periods two and three, resulting in a greater rate of water absorption compared with its excretion rate at 08:00 h, and water loss between the 2 blood collection times. However, this difference occurred even in wk 1 of period one when water was consumed ad libitum and available all day. This finding was likely due to the absorption of solutes, including volatile fatty acids and Na^+^, from the rumen after feeding [45], as feed was offered at 08:00 h. Plasma osmotic pressure usually increases 4–6 h postprandially compared with preprandial values [46]. There was not a time difference in wk 9 during the last week of period three, perhaps because of adaptive changes for water conservation. Similar OSM between periods one and three (i.e., final weeks) at 14:00 h is in accordance with this postulate, although this is not clearly the case for samples collected at 08:00 h given the three-way interaction and overall means of 302.1 and 304.5 mosmol/kg for period one and 3, respectively (SEM = 0.515). Water restriction usually results in increased ruminal as well as plasma osmotic pressure [34,47]. In period three, drinking water was offered 30 min before blood collection, which might not have allowed for water to have been absorbed and reach the vascular system to lower plasma osmotic pressure at 08:00 h [4]. Relatedly, a slow recovery in plasma osmolarity may occur following rehydration after 1 d of water restriction to prevent a sudden decline in plasma dilution [4]. In regard to this interaction for the 08:00 h time, it would also appear that there was much greater variation in adaptive processes associated with the region to conserve water for KAT than for DOR or STC. Overall, OSM was generally lower in wk 4 vs. wk 3 (period two) and wk 9 vs. wk 8 (period three), indicating that all breeds of sheep were physiologically more adapted to water restriction after the second week at a given water restriction level.

The interactions in PCV, Hb concentration, oxygen saturation of Hb, and blood oxygen concentration, particularly three-way, make the presentation of clear interpretation difficult. Hence, the main effect mean or means for two-way interactions between breed and period will be addressed for a general description of the findings. The two-way interaction (breed × period) means at 08:00 h for PCV were 31.9, 32.2, and 32.7% in period one and 31.5, 31.3, and 34.5% in period three for DOR, KAT, and STC, respectively (SEM = 0.56). Likewise, means for Hb at 14:00 h were 11.8, 11.7, and 11.7 g/dL in period one and 12.2, 11.8, and 12.4 g/dL in period three for DOR, KAT, and STC, respectively (SEM = 0.19). Furthermore, STC from different regions showed either similar or greater levels of PCV and Hb in period three compared with period one, whereas DOR and KAT from some regions had lower levels of these variables in period three than in period one. This may relate to generally lower feed intake in period three vs. 1 for DOR and KAT in contrast to fairly similar values for STC [21]. From this study, it appeared that STC with restricted water availability was able to incur a higher concentration of blood solids relative to DOR and KAT.

Although there was a breed × region interaction in oxygen saturation of Hb at 08:00 h, the main effect mean for STC was greater than for DOR and KAT (68.0, 67.4, and 74.1% for DOR, KAT, and STC, respectively; SEM = 1.55). This is in agreement with saturation at 14:00 h. Hence, regardless of water availability, STC was able to more highly saturate Hb with oxygen. As a result, the blood oxygen concentration was highest among breeds for STC, particularly at 14:00 h. Two-way interaction means at 08:00 h were 4.86, 4.93, and 5.25 mmol/L in period one and 4.89, 4.81, and 5.74 mmol/L in period three for DOR, KAT, and STC, respectively; SEM = 0.160). The concentration at 14:00 h was 4.37, 4.61, and 4.74 mmol/L in period one and 4.66, 4.81, and 5.46 mmol/L n period three for DOR, KAT, and STC, respectively (SEM = 0.154). Hence, by some means, STC was more capable of oxygenating blood with restricted water availability relative to the other breeds. The oxygen saturation of Hb depends on pH, temperature, carbon dioxide, and 2,3-bisphosphate, with lower temperature and higher pH enhancing the affinity of oxygen to Hb [48]. It seems that STC was better able to maintain higher blood pH and lower carbon dioxide concentration in blood despite water deprivation compared with other breeds. In this regard, Tadesse et al. [29] noted that with high heat load conditions, resilience to heat stress ranked STC > KAT > DOR. It was postulated that this, at least in part, related to the breaths of STC being the deepest, with the greatest volume of air being moved per breath. This suggests that physiological conditions differing among breeds and impacting resilience to stress factors could be relevant to multiple types of stressors.

### 4.2. Blood Constituent Concentrations—Wk 2, 4, 6, 8, and 9

#### 4.2.1. Period

Water restriction has had varying effects on blood glucose concentrations in sheep, with either a decrease [49] or no change being observed [8,9,20]. Because ingested carbohydrates are efficiently fermented to volatile fatty acids in the rumen, circulating glucose in ruminants is derived from propionate and other non-carbohydrate precursors (e.g., lactate, glycerol, and amino acids) through gluconeogenesis [50]. In situations where water restriction causes decreases in feed intake, propionate production in the rumen can also decrease and lead to reductions in blood glucose concentrations [51]. As a key gluconeogenic substrate, circulating lactate is an end product of fermentation in the gastrointestinal tract or nonoxidative glycolysis in tissues [52]. In either case, lactate concentration could also be influenced by water restriction because up to 70% of circulating lactate is subject to renal tubular reabsorption [53].

As expected, all blood constituents in the current experiment were affected by week depending on the severity and adaptation to water restriction. Factors responsible for decreased levels of glucose and lactate from wk 2 to 4 are unclear, but slightly lower feed intake with the 25% restriction in available drinking water in period two may have been involved [21]. Water restriction in animals can increase cortisol concentrations in blood [22,34], although this was not observed in the present experiment and reflects adaptation mechanisms to water restriction. Lactate is a product of glycogenolysis and nonoxidative glycolysis processes that are activated by high energy needs in muscle [52], and a major portion of circulating lactate (up to 70%) is reabsorbed in renal tubules [51]. That blood lactate concentration was not markedly affected by water restriction, although hemoconcentration might indicate lower energy metabolism in muscle due to reduced activity.

Urea is mainly synthesized in the liver from ammonia, released to the blood, and excreted by the kidneys to dispose of endogenous and excess dietary N or recycled through saliva and reabsorption into the rumen to be utilized by rumen bacteria [54]. Creatinine is produced in muscles and excreted by the kidneys in proportion to muscle mass and the rate of proteolysis [55]. However, with water restriction, the transfer function of the kidney is altered [16], resulting in slower glomerular filtration and higher reabsorption [25,56]. Plasma urea N concentration has a high negative correlation with glomerular filtration rate, and urea N in the urine primarily originates from filtrate urea N with a substantial amount of urea secretion in the tubules of nephrons by urea transporters [57]. Furthermore, a significant proportion of filtrate urea can be reabsorbed in the distal tubules of nephrons [57]. As a result, blood concentrations of urea and creatinine have been increased in sheep by water restriction [8,9,23], although the results of this experiment are not in close accordance with these findings. In this regard, the pattern of change in urea N concentration with advancing time was unique relative to most other constituents, with an increasing concentration to wk 6 and 8 and then a decline to wk 9. The decreased plasma urea N concentration in wk 9 compared with the values at wk 6 and 8 might be attributed to increased urea secretion into kidney tubules by greater urea transporter activities and lowered urea reabsorption from the tubules as a result of physiological adaptation to long-term dehydration. Previous studies reported that urea secretion is increased during water restriction conditions [57,58]. Similar to the urea N excretion mechanism in the kidney tubules, filtrate creatinine and secreted creatinine mostly contribute to urinary creatinine with some amount of creatinine reabsorption [59,60]. The increasingly lower concentrations of plasma creatinine after wk 6 may also relate to a greater secretion rate than the reabsorption rate in the tubules. In this context, it is well known that urea reabsorption is usually greater than creatinine reabsorption in the tubules [59]. This may be related to increasing concentrations of plasma urea N from wk 4 to wk 8, while creatinine concentration increased in wk 6 and gradually decreased thereafter. The decreasing creatinine concentration wk 6 to 9 may also reflect adaptation through decreasing tissue protein mobilization.

According to Caldeira et al. [55,61], serum concentrations of total protein and albumin are predictive of animal protein status, and a decrease in albumin concentration is common in ruminants suffering from prolonged low dietary protein intake. This decrease is because serum albumin serves as a labile protein reservoir [62]. Because albumin plays an important role in osmoregulation and control of fluid movement between different body compartments, its breakdown and synthesis are regulated in response to dehydration to maintain normal colloid osmotic pressure and fluid distribution [63]. Considering these physiological functions, reported decreases in concentrations of total protein and albumin in sheep have resulted from limited feed intake caused by water restriction [2,64]. However, increases in concentrations of total protein and albumin have also been reported in water-restricted sheep [7,8,20], being attributed to decreased blood volume [41]. Findings for total protein and albumin in this experiment were fairly similar to those for glucose and lactate, with lower values in wk 4 vs. 2 but concentrations in wk 9 of period three similar to those in period one. For each of these constituents, perhaps the levels in wk 6 with the greatest water restriction level were not lower than in wk 4 with moderate restriction involving greater adaptive physiological changes than had occurred in the first 2 wk with the initial restriction level.

Concentrations of cholesterol and triglycerides are also considered reflective of drinking water availability. Increased serum concentrations of cholesterol [8,9,65] and triglycerides [20,66] in water-restricted sheep have been attributed to decreased feed intake and the subsequent need for fat mobilization to meet the shortfall in energy supply [67]. In accordance, feed intake was slightly decreased by water restriction in this study [21]. However, generally greater levels of cholesterol and triglycerides in period three than earlier could relate to increased digestibility and BW during water restriction [21,26]. It was also reported that complete water restriction for 4 d reduced thyroid hormone concentration in the blood of sheep [22], which has been negatively related to blood cholesterol and triglyceride levels [68].

#### 4.2.2. Breed

The lower blood glucose concentration for DOR than for STC is somewhat in accordance with the findings of Tadesse et al. [27] involving responses to high heat load conditions, with the lowest value among breeds for DOR. Although, in a feed restriction study [28], blood glucose concentration was similar among these breeds. The similar concentration of lactate among breeds agrees with findings from studies addressing the effects of limited feed intake [28] and high heat load conditions [27]. The highest concentration of urea N among breeds for STC in the current experiment is in accordance with these two other studies as well. As in these studies, the breed ranking in creatinine presumably reflects differences in muscle mass [27,28]. The lowest body condition score and level of fatness for STC could also account for the relatively low concentration of triglycerides [27,28].

### 4.3. Blood Constituent Concentrations—Wk 2 and 9

Although means in the last week of periods one and three (i.e., wk 2 and 9) have been described earlier, there are some differences in effects worth addressing. One is the creatinine concentration, with a three-way interaction involving breed, region, and period. The concentration of creatinine in STC from all regions was similar in wk 2 of period one and wk 9 of period three, whereas creatinine concentration in DOR from NW and KAT from MW was still greater in wk 9 vs. wk 2. This suggests an excellent capacity for adaptation of STC from all regions, converse to DOR and KAT from some regions that displayed less adaptability.

Given the function of the renin–angiotensin–aldosterone system in regulating blood pressure and levels of sodium and potassium, the concentration of aldosterone was expected to be altered with restricted water intake. However, its concentration was similar between periods one and three. This may suggest that these breeds were able to maintain plasma osmotic pressure and electrolyte balance during water restriction without altering aldosterone concentration. In other studies, water restriction did not alter aldosterone concentrations in goats and sheep [18,34] though it increased plasma rennin activity [18]. Heat-stress-related dehydration reduced aldosterone concentration in Malabari goats but not in Osmanabadi and Salem Black goats. It has been suggested that Osmanabadi and Salem goats have excellent capability to adapt to harsh conditions [15]. In Bedouin goats that have extreme resilience to severe water deprivation, aldosterone concentration along with rennin activity increased during dehydration conditions [19]. Similar to the marked difference in aldosterone concentration between STC and DOR and KAT in the present experiment, a substantial breed difference was noted in another study, with Malabari and Salem Black goats having a plasma aldosterone concentration twice that in Osmanabadi goats [15]. The similar concentration of cortisol between periods in the current experiment may have been a function of the many adaptive physiological changes that prevented or minimized stress after the periods and weeks of restricted water availability compared with ad libitum intake in period one.

Stress factors, including water restriction, activate the hypothalamus-pituitary-adrenal axis, resulting in the secretion of cortisol [34,69]. In sheep, limiting the time of access to water to 3 h/d increased the concentration of plasma cortisol [17]. Water availability below 60% of ad libitum intake caused an increase in cortisol concentration in blood [34]. In the present study, the similar concentration of cortisol in wk 2 and wk 9 suggests that these breeds experienced minimal to no stress elicited by a 50% water restriction after long-term adaptation. Water restriction for 24 h also did not alter cortisol concentration in beef heifers [24]. Hussein et al. [21] concluded that based on BW and feed intake, STC was more consistent in the display of high resilience to restricted drinking water availability compared with DOR and KAT. The much greater concentration of aldosterone for STC vs. DOR may have contributed to this difference in resilience in some manner. Overall, all blood metabolites in wk 6 with the greatest water restriction were altered, but most of these blood metabolites except urea N, triglycerides, and cholesterol had returned to values similar to those observed during ad libitum water intake, suggesting resilience of these breeds to water restriction up to 50%.

## 5. Conclusions and Implications

There were relatively few effects of the region on blood characteristics or constituent levels measured. Despite the substantial difference in drinking water availability, in general, the sheep breeds used in this study displayed considerable adaptation capacity. Among the three breeds, St. Croix showed better adaptability from all four regions as they maintained osmotic pressure during water restriction conditions, which was not true for Dorper and Katahdin from some regions. St. Croix also had more highly oxygenated hemoglobin and blood oxygen concentration than Dorper and Katahdin. Collectively, blood characteristics and constituents were marginally altered or unaffected by daily water restriction up to 50% of ad libitum intake, with the least changes in St. Croix and concentrations of many constituents returned to the baseline values during ad libitum water intake. This study suggests that these sheep breeds have excellent adaptability in maintaining water and electrolyte balance during long periods of limited water availability, and therefore, these breeds of hair sheep, especially St. Croix, may be recommended for farming in arid regions of the world, including in the USA.

## Figures and Tables

**Figure 1 animals-12-03167-f001:**
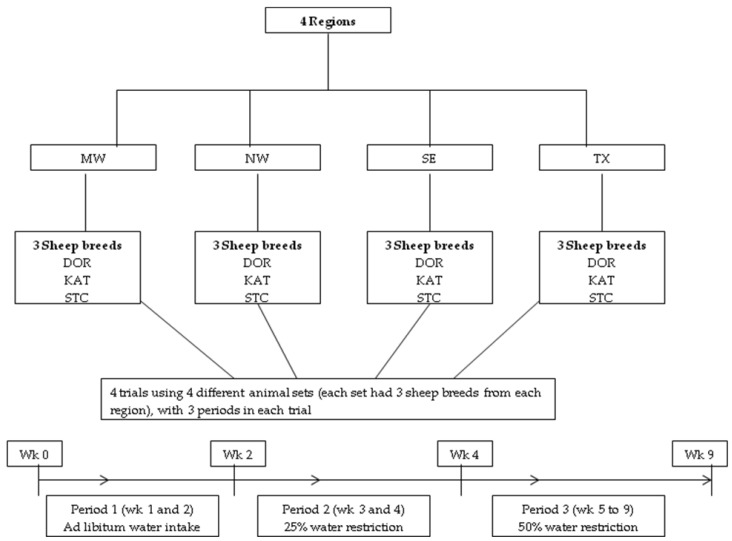
A schematic presentation of the experimental design to evaluate effects of water restriction on 3 sheep breeds, Dorper (DOR), Katahdin (KAT), and St. Croix (STC), from 4 different climatic regions, i.e., Midwest (MW), Northwest (NE), Southeast (SE), and central Texas (TX) of the USA.

**Table 3 animals-12-03167-t003:** Effects of hair sheep breed, region of origin, period with different levels of drinking water availability, and week within periods one and two and the last 2 wk of period three on packed cell volume, hemoglobin concentration and oxygen (O_2_) saturation, and blood O_2_ concentration at 08:00 h in 3 breeds of hair sheep from 4 different regions of the USA ^1,2^.

			Dorper^3^					Katahdin					St. Croix				
Variable ^4^	Prd ^5^	Wk ^5^	MW	NW	SE	TX		MW	NW	SE	TX		MW	NW	SE	TX	SEM
PCV (%)	1	1	35.3	31.8	31.8	32.7		32.0	32.5	33.0	29.9		33.2	32.7	31.0	32.7	1.08
	1	2	31.2	32.7	31.0	32.9		33.4	31.8	32.0	32.0		33.0	32.6	31.4	33.1	
	2	3	33.2	30.9	30.4	33.8		33.1	30.7	33.2	32.7		33.9	33.3	32.7	35.7	
	2	4	33.7	29.4	30.6	33.6		31.7	31.2	32.5	33.1		33.2	33.3	32.5	34.1	
	3	8	32.2	29.2	33.0	33.8		33.0	32.1	32.5	32.7		34.1	34.7	34.4	34.5	
	3	9	32.5	28.6	31.5	33.5		30.5	32.0	33.2	29.8		34.4	32.7	35.3	35.1	
Hb (g/dL)	1	1	13.2	12.1	12.1	12.2		12.2	12.4	12.4	11.4		12.3	12.2	11.3	11.9	0.42
	1	2	12.0	12.1	11.7	12.6		12.5	12.0	12.2	12.2		12.2	12.0	11.6	11.3	
	2	3	12.6	11.8	11.8	12.6		12.7	11.7	12.7	12.4		12.6	12.3	12.2	12.4	
	2	4	12.7	11.6	11.6	12.5		12.5	11.7	12.4	12.7		12.6	12.5	12.1	12.6	
	3	8	12.5	10.6	12.5	12.7		12.6	12.0	12.2	12.3		12.8	12.8	12.8	12.7	
	3	9	12.5	11.1	12.1	12.9		12.0	12.2	12.5	11.4		12.7	12.4	12.9	13.0	
Hb O_2_ (%)	1	1	72.0	69.7	59.7	59.2		64.8	65.1	71.4	66.1		75.8	79.5	78.4	67.3	4.24
	1	2	64.6	77.7	70.6	59.5		66.5	58.9	73.2	74.5		73.6	72.0	73.0	71.2	
	2	3	65.1	75.1	67.4	57.6		66.4	74.3	62.4	65.4		76.8	74.0	74.9	74.9	
	2	4	65.8	71.2	64.7	64.9		65.3	60.8	66.0	54.9		71.3	73.9	70.5	73.7	
	3	8	70.8	74.1	67.7	64.8		68.2	66.0	71.9	70.1		73.0	70.4	71.5	72.6	
	3	9	68.4	70.9	77.5	57.2		74.1	58.8	64.9	70.9		69.9	77.8	76.1	78.3	
O_2_ (mmol/L)	1	1	5.66	4.96	4.42	4.38		4.87	4.82	5.31	4.51		5.58	5.94	5.27	4.82	0.317
	1	2	4.60	5.63	4.82	4.51		5.00	4.06	5.36	5.45		5.40	5.18	5.04	5.27	
	2	3	4.91	5.27	4.78	4.33		5.00	5.18	4.82	4.96		5.80	5.45	5.49	5.89	
	2	4	5.00	4.96	4.55	4.91		4.91	4.24	4.91	4.24		5.31	5.49	5.09	5.49	
	3	8	5.27	5.09	5.00	4.91		5.13	4.73	5.27	5.13		5.58	5.54	5.49	5.54	
	3	9	5.09	4.64	5.45	4.46		5.36	4.33	4.91	4.78		5.31	5.71	5.85	5.03	

^1^ Water consumption was ad libitum in period one and that offered in periods two and three was 75 and 50% of ad libitum consumption. ^2^ Water was offered daily at 07:30 h. ^3^ MW = Midwest; NW = Northwest; SE = Southeast; TX = central Texas. ^4^ PCV = packed cell volume; Hb = hemoglobin. ^5^ Prd = period; Wk = week.

**Table 4 animals-12-03167-t004:** Effects of period with different levels of drinking water availability and week within periods one and two and the last 2 wk of period three on plasma osmolality (OSM), packed cell volume (PCV), and hemoglobin (Hb) concentration and oxygen (O_2_) saturation at different sampling times in 3 breeds of hair sheep from 4 different regions of the USA ^1^.

		Period 1			Period 2			Period 3		
Item	Time (h)	Wk 1	Wk 2		Wk 3	Wk 4		Wk 8	Wk 9	SEM
OSM (mosmol/kg)	08:00	298.3 ^a^	302.1 ^b^		304.7 ^c^	301.6 ^b^		305.8 ^c^	304.4 ^c^	0.63
	14:00	303.0 ^a^	302.9 ^a^		307.0 ^bc^	305.4 ^b^		307.3 ^c^	302.3 ^a^	0.65
PCV (%)	14:00	31.0 ^a^	31.3 ^ab^		32.0 ^cd^	31.1 ^ab^		31.7 ^bc^	32.2 ^d^	0.30
Hb (g/dL)	14:00	11.8 ^ab^	11.7 ^a^		12.1 ^cd^	11.9 ^abc^		12.0 ^bc^	12.2 ^d^	0.11
Hb O_2_ (%)	14:00	67.3 ^bc^	63.9 ^a^		65.2 ^ab^	66.6 ^abc^		73.8 ^d^	68.4 ^c^	1.33

^a, b, c, d^ Means without a common superscript letter within a row differ (*p* < 0.05).

**Table 8 animals-12-03167-t008:** *p* values for effects of week with water offered at 100, 75, or 50% of ad libitum intake on blood metabolite concentrations in 3 breeds of hair sheep from 4 different regions of the USA.

Source of Variation ^2^	Variable ^1^							
	GLC (mg/dL)	LAC (mg/dL)	UN (mg/dL)	TP (g/dL)	ALB (g/dL)	CRT (mg/dL)	CHL (mg/dL)	TG (mg/dL)
Set	0.030	<0.001	0.024	<0.001	<0.001	0.068	0.001	<0.001
Age	0.164	0.199	0.100	0.114	0.008	0.219	0.878	0.731
Brd	0.018	0.638	0.010	0.368	0.920	<0.001	0.235	0.001
Reg	0.980	0.219	0.481	0.162	0.473	0.413	0.550	0.656
Brd *Reg	0.612	0.818	0.509	0.072	0.256	0.025	0.523	0.177
Wk ^3^	<0.001	<0.001	<0.001	<0.001	<0.001	<0.001	<0.001	<0.001
Brd *Wk	0.894	0.696	0.753	0.910	0.756	0.774	0.659	0.745
Reg *Wk	0.051	0.183	0.199	0.994	0.951	0.564	0.553	0.503
Brd *Reg *Wk	0.228	0.187	0.659	0.876	0.773	0.134	0.894	0.854

^1^ GLC = glucose; LAC = lactate; UN = urea nitrogen; TP = total protein; ALB = albumin; CRT = creatinine; CHL = cholesterol; TG = triglycerides. ^2^ Brd = breed; Reg = region; Wk = week. ^3^ Samples were collected in wk 2, 4, 6, 8, and 9; water intake was ad libitum in wk 2, 75% of ad libitum intake in wk 4, and 50% of ad libitum intake in wk 6, 8, and 9. * It indicates the interaction effects of the factors.

**Table 9 animals-12-03167-t009:** Effects of week with water offered at 100, 75, or 50% of ad libitum intake on blood metabolite concentrations in 3 breeds of hair sheep from 4 different regions of the USA.

		Breed ^1^				Region ^2^					Week ^3^					
Item ^4^	Brd ^5^	DOR	KAT	STC	SEM	MW	NW	SE	TX	SEM	2	4	6	8	9	SEM
GLC		53.7 ^a^	55.4 ^ab^	57.6 ^b^	0.94	55.6	55.5	56.0	55.3	1.08	55.5 ^b^	51.0 ^a^	59.0 ^c^	56.2 ^b^	56.3 ^b^	0.91
LAC		22.0	21.9	22.8	0.68	22.1	23.5	22.2	21.2	0.77	24.0 ^c^	20.5 ^a^	21.6 ^ab^	22.8 ^bc^	22.3 ^abc^	0.73
UN		20.6 ^a^	20.7 ^a^	22.4 ^b^	0.44	21.1	21.0	21.9	20.9	0.50	19.4 ^a^	20.8 ^b^	22.5 ^c^	22.5 ^c^	21.0 ^b^	0.38
TP		7.07	7.07	7.22	0.084	7.05	7.23	7.23	6.98	0.098	7.08 ^b^	6.62 ^a^	7.29 ^bc^	7.43 ^c^	7.19 ^b^	0.084
ALB		2.63	2.62	2.61	0.029	2.62	2.64	2.65	2.58	0.033	2.59 ^b^	2.43 ^a^	2.68 ^cd^	2.74 ^d^	2.65 ^bc^	0.029
CRT											0.840 ^ab^	0.815 ^a^	0.959 ^d^	0.919 ^c^	0.864 ^b^	0.0131
	DOR					0.970 ^c^	1.070 ^d^	0.944 ^c^	0.939 ^c^	0.0331						
	KAT					0.860 ^bc^	0.915 ^c^	0.907 ^c^	0.845 ^b^							
	STC					0.773 ^ab^	0.730^a^	0.787^ab^	0.811^ab^							
CHL		64.7	67.3	63.1	1.72	66.9	62.8	65.2	65.4	1.99	56.2 ^a^	57.3 ^a^	69.5 ^b^	72.9 ^c^	69.3 ^b^	1.33
TG		32.3 ^b^	35.1 ^b^	28.3 ^a^	1.24	31.7	33.4	31.7	30.9	1.44	28.6 ^a^	28.3 ^a^	33.7 ^b^	34.5 ^b^	34.5 ^b^	0.98

^a,b,c,d^ Means within grouping without a common superscript letter differ (*p* < 0.05). ^1^ DOR = Dorper; KAT = Katahdin; STC = St. Croix. ^2^ MW = Midwest; NW = Northwest; SE = Southeast; TX = central Texas. ^3^ Water consumption was ad libitum in wk 2, 75% of ad libitum consumption in wk 4, and 50% of ad libitum consumption in wk 6, 8, and 9. ^4^ GLC = glucose (mg/dL); LAC = lactate (mg/dL); UN = urea nitrogen (mg/dL); TP = total protein (g/dL); ALB = albumin (g/dL); CRT = creatinine (mg/dL); CHL = cholesterol (mg/dL); TG = triglycerides (mg/dL). ^5^ Brd = breed.

**Table 10 animals-12-03167-t010:** *p* values for effects of period with water offered at 100 and 50% of ad libitum intake (periods one and three, respectively) on blood metabolite concentrations in samples collected in the last week of periods in 3 hair sheep breeds from 4 different regions of the USA.

Item ^2^	Source of Variation ^1^								
	Set	Age	Brd	Reg	Brd *Reg	Prd^2^	Brd *Prd	Reg *Prd	Brd *Reg *Prd
GLC (mg/dL)	0.010	0.512	0.032	0.937	0.698	0.511	0.209	0.931	0.873
LAC (mg/dL)	<0.001	0.772	0.795	0.107	0.946	0.153	0.259	0.261	0.098
UN (mg/dL)	0.085	0.174	0.001	0.727	0.683	0.001	0.981	0.059	0.324
TP (g/L)	0.368	0.037	0.243	0.085	0.225	0.357	0.613	0.851	0.928
ALB (g/L)	0.017	0.078	0.929	0.516	0.348	0.090	0.556	0.813	0.946
CRT (mg/dL)	0.385	0.338	<0.001	0.363	0.020	0.070	0.297	0.134	0.004
CHL (mg/dL)	0.005	0.860	0.371	0.965	0.817	<0.001	0.469	0.325	0.714
TG (mg/dL)	<0.001	0.575	<0.001	0.179	0.153	<0.001	0.222	0.777	0.883
COR (ng/mL)	0.630	0.699	0.882	0.744	0.867	0.324	0.223	0.513	0.334
ALD (pg/mL)	0.150	0.831	0.008	0.881	0.252	0.312	0.524	0.274	0.185

^1^ Brd = breed; Reg = region; Prd = period. ^2^ GLC = glucose; LAC = lactate; UN = urea nitrogen; TP = total protein; CRT = creatinine; CHL = cholesterol; TG = triglycerides; COR = cortisol; ALD = aldosterone. * It indicates the interaction effects of the factors.

**Table 11 animals-12-03167-t011:** Effects of period with water offered at 100 and 50% of ad libitum intake (periods one and three, respectively) on blood metabolite concentrations in samples collected in the last week of periods in 3 hair sheep breeds from 4 different regions of the USA.

			Breed ^1^				Region ^2^					Prd ^3^		
Item ^4^	Brd ^5^	Prd	DOR	KAT	STC	SEM	MW	NW	SE	TX	SEM	1	3	SEM
GLC			53.8 ^a^	55.3 ^ab^	58.0 ^b^	1.10	56.3	55.5	55.7	55.2	1.26	55.3	56.1	0.84
LAC			22.9	23.3	23.8	0.98	22.5	25.6	23.4	21.8	1.11	24.1	22.5	0.79
UN			19.2 ^a^	19.7 ^a^	21.6 ^b^	0.46	19.8	20.1	20.6	20.1	0.53	19.4 ^a^	21.0 ^b^	0.34
TP			7.04	7.06	7.27	0.102	6.94	7.32	7.23	7.01	0.117	7.08	7.17	0.078
ALB			2.63	2.61	2.62	0.039	2.58	2.65	2.66	2.59	0.044	2.59	2.65	0.029
CRT	DOR	1					0.991^y^	0.961 ^x^	0.901	0.890	0.0421			
	DOR	3					0.882^x^	1.118 ^y^	0.889	0.900				
	KAT	1					0.761 ^x,a^	0.865 ^ab^	0.879 ^b^	0.826 ^ab^				
	KAT	3					0.914 ^y,b^	0.941 ^b^	0.896 ^ab^	0.801 ^a^				
	STC	1					0.761	0.699	0.777	0.756				
	STC	3					0.742	0.704	0.764	0.818				
CHL			64.7	67.3	63.1	1.72	66.9	62.8	65.2	65.4	1.99	56.2 ^a^	69.3 ^b^	1.33
TG			32.3 ^b^	35.1 ^b^	28.3 ^a^	1.24	31.7	33.4	31.7	30.9	1.44	28.6 ^a^	34.5 ^b^	0.98
COR			6.43	6.98	6.65	0.787	5.94	7.31	6.53	6.97	0.907	7.02	6.36	0.560
ALD			52.0 ^a^	62.5 ^a^	108.4 ^b^	13.07	78.8	80.8	65.5	72.0	15.08	78.2	70.3	8.45

^x,y^ Period means within breed × region grouping with a different superscript letter differ (*p* < 0.05). ^a,b,c,d^ Means within grouping without a common superscript letter differ (*p* < 0.05). ^1^ DOR = Dorper; KAT = Katahdin; STC = St. Croix. ^2^ MW = Midwest; NW = Northwest; SE = Southeast; TX = central Texas. ^3^ Water consumption was ad libitum in period one and 50% of ad libitum consumption in period three. ^4^ GLC = glucose (mg/dL); LAC = lactate (mg/dL); UN = urea nitrogen (mg/dL); TP = total protein (g/dL); ALB = albumin (g/dL); CRT = creatinine (mg/dL); CHL = cholesterol (mg/dL); TG = triglycerides (mg/dL); COR = cortisol (ng/mL); ALD = aldosterone (pg/mL). ^5^ Brd = breed.

## Data Availability

Mean data are presented in tables.

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
