# Peer review of "Effects of Restricted Availability of Drinking Water on Blood Characteristics and Constituents in Dorper, Katahdin, and St. Croix Sheep from Different Regions of the USA"

_animals, 2022, doi:10.3390/ani12223167_

Round 1

Reviewer 1 Report

Manuscript ID: animals-2003696
Title: Effects of Restricted Availability of Drinking Water on Blood Characteristics and Constituents in Dorper, Katahdin, and St. Croix Sheep from Different Regions of the USA
Authors: Ali Hussein Hussein, Amlan Kumar Patra, Ryszard Puchala, Blake Kenyon Wilson, and Arthur Louis Goetsch

In this work the metabolic and hormonal responses of 3 fiber sheep breeds from 4 regions and for 4 periods to the restriction of water during the day are evaluated.

The problem addressed is current and interesting, however the many tables do not allow an effective reading of the results. I would suggest to the authors, in order to make the reading of the significant results in the tables more immediate, to omit the not significant ones from the P value tables. The latter can be inserted into the work as additional material at the end of the work.

Line 87-90: It would be interesting to know of the animals used for experimentation also the Body Condition Score, also in the light of the discussion in line 470-471

Author Response

In this work the metabolic and hormonal responses of 3 fiber sheep breeds from 4 regions and for 4 periods to the restriction of water during the day are evaluated.

The problem addressed is current and interesting, however the many tables do not allow an effective reading of the results. I would suggest to the authors, in order to make the reading of the significant results in the tables more immediate, to omit the not significant ones from the P value tables. The latter can be inserted into the work as additional material at the end of the work.

Response: We preferred to present the p-value tables close to those presenting mean value results so that readers can understand the presentation of data results properly and why we did not present interaction effects for all variables. We presented the interaction effects when they were significant. We hope the reviewer could understand our explanations.

Line 87-90: It would be interesting to know of the animals used for experimentation also the Body Condition Score, also in the light of the discussion in line 470-471.

Response: This is a good suggestion, but we did not measure BCS in this study. We explained the results obtained in this paper based on other studies.

Reviewer 2 Report

Dear authors, it seems to me that this manuscript has great relevance in the scientific world. However, many points affect the quality of the manuscript.

Correct the language and writing style. Some lines are difficult to read.

General comments:

Abstract

Add P-values. 

Please write the objective of the manuscript as a separated text (E.g. The objective is ..... in sheep. For this study 40 dorper ..... were used.). 

The description of the results here is complete; however, I see a lot of information, some of which is not detailed in the objective of the manuscript, so I think that the objective of the abstract should be rewritten.

The conclusion here is very generic: (blood variables depending on the severity and duration of the restriction) OK and? I think everyone expected that result. Remember, the abstract is the document that first represents your manuscript.

Introduction

This is very well written; however, only lines 80-81 are the differential of your manuscript? What does “constituent levels” mean? This is a characterization of the races with water deprivation? If the study is a characterization, why didn't you investigate more breeds?

Studies in arid or semi-arid regions have already been done in other countries and with these breeds, what is different in your study?

Conclusion:

I recommend separating this into two topics:Conclusion and Implications. I don't recommend a new summary because there is an “abstract”and a simple summary.

References: Please update references. Many references are very old. Use ten years as the reference date, considering a maximum of 20% of the references with a date greater than ten years.

Specific comments:

Line 98:  Add here in parentheses “Figure 1” and delete line 103.

Lines 103-107: If the study is about water stress of sheep raised in arid or semi-arid conditions, it is necessary to add the descriptions of the region and the climate. Parameters such as: rain, temperature, wind speed, humidity, altitude, latitude, etc.

Line 118: Please present energy on a digestible or metabolizable basis.

Lines 118-120: What is the purpose of these lines?Suggestion: remove.

Line 119: What forage?

Line 122: If the objective is water restriction in a semi-arid or arid climate; why was the blood collection not done in another hour? It's just a question

Lines 139-141What is the purpose of these lines?Suggestion: remove.

Lines 149-152: I think that your objective was to write to made a clear text but to me it was confuse. Furthermore, why two different hours? This is part of the statistical analysis?

Lines 149-169: It is not clear the number of samples collected, the number of samples used for each analysis, the volume collected, the volume used for each analysis, etc.

Lines 171-174: What is the purpose of these lines?Suggestion: remove.

Tables 6 and 7, and other tables are not ideally formatted.

Author Response

Dear authors, it seems to me that this manuscript has great relevance in the scientific world. However, many points affect the quality of the manuscript. Correct the language and writing style. Some lines are difficult to read.

Response: Thanks for your useful suggestions. We have revised the manuscript following your suggestions.

General comments:

Abstract: 

Add P-values. 

Response: some p-values are added.

Please write the objective of the manuscript as a separated text (E.g. The objective is ..... in sheep. For this study 40 dorper ..... were used.). 

Response: This has been revised as per the suggestion.

The description of the results here is complete; however, I see a lot of information, some of which is not detailed in the objective of the manuscript, so I think that the objective of the abstract should be rewritten.

Response: The objective of the study has been added/revised.

The conclusion here is very generic: (blood variables depending on the severity and duration of the restriction) OK and? I think everyone expected that result. Remember, the abstract is the document that first represents your manuscript.

Response: The conclusion has been revised.

Introduction: 

This is very well written; however, only lines 80-81 are the differential of your manuscript? What does “constituent levels” mean? This is a characterization of the races with water deprivation? If the study is a characterization, why didn't you investigate more breeds?

Studies in arid or semi-arid regions have already been done in other countries and with these breeds, what is different in your study?

Response: In the L80-81, we have now mentioned “biochemical concentrations” instead of “constituents levels.” The main purpose of this study was to assess how three important hair breeds of sheep raised in different regions of the USA adapted to the typical ecological conditions over the years and if there were breed differences among the regions. 

Conclusion:

I recommend separating this into two topics: Conclusion and Implications. I don't recommend a new “summary” because there is an “abstract” and a “simple summary”.

 Response: We have revised this section as per the comments of the reviewer.

References: Please update references. Many references are very old. Use ten years as the reference date, considering a maximum of 20% of the references with a date greater than ten years.

Response: About 30% of references used in the manuscript is within the last 10 years.

Specific comments:

Line 98:  Add here in parentheses “Figure 1” and delete line 103.

Response: “Figure 1” has been added in parenthesis in the next sentence.

Lines 103-107: If the study is about water stress of sheep raised in arid or semi-arid conditions, it is necessary to add the descriptions of the region and the climate. Parameters such as: rain, temperature, wind speed, humidity, altitude, latitude, etc.

Response: We have provided some information about the climatic conditions of these regions.  Because of the distribution and number of animals within these regions and variability among geographic location and time in environmental conditions, the ability to present clear, concise, and accurate descriptions over appreciable lengths of time seems limited.  However, as the locations are described, readers can search various sources such as internet sites for specific information if so desired.  In this regard, NEON Domains and National Centers for Environmental Information have now been listed.

Line 118: Please present energy on a digestible or metabolizable basis.

Response: Some information about assumptions are presented, although more about digestibility and metabolizability would be available from Hussein et al. (2020, 10, 10032), indicating that metabolizability can vary with water restriction.

Lines 118-120: What is the purpose of these lines? Suggestion: remove.

Response: This sentence has been deleted as per the suggestion.

Line 119: What forage?

Response: the sentence related to it is deleted now.

Line 122: If the objective is water restriction in a semi-arid or arid climate; why was the blood collection not done in another hour? It's just a question

Response: The question is not clear to us. It was written “Feed was offered twice daily at 08:00 and 15:00 h, except for the morning meal being 1 h later on Wednesday because of blood sample collection” Blood samples were collected twice weekly at 08:00 h on d 6 and 14:00 h on d 7 (i.e., 1 and 7 h in period 1 and 0.5 and 6.5 h in period 2 and 3 after drinking water was offered, respectively). This was mentioned under the subsection “Measures”.

Lines 139-141: What is the purpose of these lines? Suggestion: remove.

Response: This sentence was meant to explain why 25 and 50% water restriction was used. This information may be useful to the readers.

Lines 149-152: I think that your objective was to write to made a clear text but to me it was confuse. Furthermore, why two different hours? This is part of the statistical analysis?

Response: Two hours of collection was chosen to understand the effect on the blood chemical concentrations and characteristics after 0.5-1 h and 6.5-7 h after water intake. Yes, time was included in the statistical analysis.

Lines 149-169: It is not clear the number of samples collected, the number of samples used for each analysis, the volume collected, the volume used for each analysis, etc.

Response: Blood samples were collected from all sheep and all samples were used for analysis. The volume of the samples used was according to the protocols.

Lines 171-174: What is the purpose of these lines? Suggestion: remove.

Response: We have revised this part.

Tables 6 and 7, and other tables are not ideally formatted.

Response: The tables have been formatted. However, they are sometimes changed by the Editorial office to fit within its desired formats.

Reviewer 3 Report

An important aspect of the water reduction for sheep was taken up in the paper.

Comments

  1.  There is no purpose in the abstract, no clearly described research hypothesis
  2. Correctly written introduction, please justify the selection of the sheep breed for the research
  3. The material of the method please complete:

-         please add whether the sheep were lactating or barren, at what age were the lambs:

-        please describe in detail the maintenance conditions, temperature, humidity, air exchange in each experiment (zoohygienic conditions)

  1. The results: The tables should be in the results chapter, not Statistical analysis. This disrupts the publication layout
  2. The summary should indicate the practical aspect of the research and the possibility of introducing it into practice

Author Response

There is no purpose in the abstract, no clearly described research hypothesis

Response: A hypothesis and objective of the study have been provided now.

Correctly written introduction, please justify the selection of the sheep breed for the research

Response: It was mentioned at the beginning of the last paragraph of the Introduction section. This part also has now been modified slightly.

The material of the method please complete:

-         please add whether the sheep were lactating or barren, at what age were the lambs:

-        please describe in detail the maintenance conditions, temperature, humidity, air exchange in each experiment (zoohygienic conditions)

Response: Age information was provided – “Forty-five Dorper [DOR; mean ± standard error; initial body weight (BW) = 60.7 ± 1.76 kg and age = 3.7 ± 0.34 yr], 45 Katahdin (KAT; 62.7 ± 1.87 kg and 3.9 ± 0.36 yr), and 44 St. Croix (STC; 44.2 ± 1.88 kg and 2.7 ± 0.29 yr) female sheep were used.  None of the sheep were lactating when procured or used in this experiment, as addressed by Hussein et al. (2019).  It has now been clarified that environmental conditions during the experiment are preseted in Husstein et al. (2019).

The results: The tables should be in the results chapter, not Statistical analysis. This disrupts the publication layout.

Response:  The tables now have been placed in the Results section.  Initial two tables had information about the statistical analyses and explained procedures for different variables for many main effects and their interactions. Therefore, these two tables have been placed in the statistics section and other tables are presented in the Results section.

The summary should indicate the practical aspect of the research and the possibility of introducing it into practice.

Comments:  We have revised this section.

Reviewer 4 Report

The idea of the manuscript is novel and worth to be studied. There are few comments I found that authors have to address to make this manuscript ready for publication. 

Lines 33-34: is there a statistical analysis on blood oxygen? If yes, please write down if there were significant differences. 

Lines 40-43: many blood parameters were not different among the treatments such as glucose, lactate, total protein .... etc as mentioned previously and I am not sure if the oxygen concentrations differed !!!!, please this statement need more clarification 

Lines 59-65: please insert references

- Tables 1, 2, 5, 8, and 9 are not needed, please delete. Authors can have them as a supplementary file 

Lines 515-533: please focus on writing the summary or conclusion not mentioning the results again in this section, please shorten this part of the manuscript 

Author Response

The idea of the manuscript is novel and worth to be studied. There are few comments I found that authors have to address to make this manuscript ready for publication. 

Comments: Thanks for your valuable suggestions.

Lines 33-34: is there a statistical analysis on blood oxygen? If yes, please write down if there were significant differences. 

Response: There was an overall breed effect (P < 0.001) on blood oxygen concentration.

Lines 40-43: many blood parameters were not different among the treatments such as glucose, lactate, total protein .... etc as mentioned previously and I am not sure if the oxygen concentrations differed !!!!, please this statement need more clarification.

Response: The breed P value on blood oxygen level has been provided for 8:00 and 14:00 h.

Lines 59-65: please insert references

Response:  Some references have been added here now.

- Tables 1, 2, 5, 8, and 9 are not needed, please delete. Authors can have them as a supplementary file.

Response: We feel this is a relatively complex statistical analysis involving several factors including interaction effects. Therefore, the statical analysis tables may be useful to the readers to understand the results properly.

Lines 515-533: please focus on writing the summary or conclusion not mentioning the results again in this section, please shorten this part of the manuscript.

Response:  We have revised and shortened this section as per the suggestions.

Round 2

Reviewer 2 Report

Dear authors, I congratulate you on the work you have done.  You made corrections to the manuscript according to my suggestions.  At this point, I only have one observation. The number of animals in abstract does not agree with the number of animals in the Materials and Methods Topic.

Specific comments:

Lines 27-28: That number of animals does not match the number of animals on lines 96-98.

Author Response

Dear authors, I congratulate you on the work you have done.  You made corrections to the manuscript according to my suggestions.  At this point, I only have one observation. The number of animals in abstract does not agree with the number of animals in the Materials and Methods Topic.

Response: Thanks again for your valuable suggestions.

Lines 27-28: That number of animals does not match the number of animals on lines 96-98.

Response: We stated in statistical section that "the data from 6 animals (1 DOR, 3 KAT and 2 STC) were removed due to the possibility of not being adequately adapted to conditions in period 1". Thus, we wanted to mention the number of animals that were actually used in the analysis in the Abstract. But now we have mentioned the number of animals that were used in the experiment to avoid confusion. We have corrected it in the Abstract of the revised manuscript.